# Tissue and Circulating MicroRNAs 378 and 142 as Biomarkers of Obesity and Its Treatment Response

**DOI:** 10.3390/ijms241713426

**Published:** 2023-08-30

**Authors:** Georgy A. Matveev, Natalya V. Khromova, German G. Zasypkin, Yulia A. Kononova, Elena Yu. Vasilyeva, Alina Yu. Babenko, Evgeny V. Shlyakhto

**Affiliations:** World-Class Scientific Center “Center for Personalized Medicine”, Almazov National Medical Research Centre, St. Petersburg 197341, Russiakhromova@almazovcentre.ru (N.V.K.); zasypkin_gg@almazovcentre.ru (G.G.Z.);

**Keywords:** miR-378, miR-142, obesity, subcutaneous adipose tissue (SAT)

## Abstract

Promising approaches to the treatment of obesity include increasing energy expenditure and slowing down fibrogenesis of adipose tissue. The neurotransmitter reuptake inhibitor sibutramine affects appetite and activates lipolysis in a catecholaminergic way. MicroRNAs (miRs) are considered as biomarkers of molecular genetic mechanisms underlying various processes. The profile of a number of miRs is altered in obesity, both in the circulation and in adipose tissue. The aim of this study was to assess the expression levels of miRs (hsa-miR-378a-3p, hsa-miR-142-3p) by real-time polymerase chain reaction in subcutaneous adipose tissue (SAT) and in plasma in patients with different degrees and duration of obesity and during sibutramine therapy. This study included 51 obese patients and 10 healthy subjects with normal weight who formed a control group. The study found that, before treatment, obese patients had no significant difference in the expression level of miR-378 in SAT and plasma compared to the control group, while the expression of miR-142 was significantly decreased in SAT and increased in plasma. A significant elevation in miR-378 expression level was noted in patients with first-degree obesity and duration of less than 10 years, and the decline in miR-142 increased with the duration of obesity. These data indicate a maximal increase in the expression of the adipogenesis inducer miR-378 in the early stages of obesity, a progressive decrease in the expression of the fibrogenesis inhibitor miR-142 in SAT with growth of duration of obesity and the likely presence of antifibrogenic effects of sibutramine realized through miR-142 activation.

## 1. Introduction

The obesity pandemic is one of the most important problems of modern public health [1] and is characterized by extremely low efficiency of treatment and catastrophic consequences for human health [2]. The most detrimental effect on health is caused by well-known visceral obesity, associated with metabolic disorders (dyslipidemia, arterial hypertension, carbohydrate metabolism disorders) and cardiovascular diseases [2]. Thus, there is a need for the identification of early markers that allow one to estimate the risk of emergence of this phenotype of obesity and, whenever possible, methods of prevention of its development. One of the mechanisms of switching the accumulation of excess energy from the subcutaneous fat depot to the visceral one is the activation of fibrosis in the subcutaneous adipose tissue with deterioration of its plasticity and limitation of further expansion [3]. Brown adipose tissue (BAT) activation is associated with inhibition of profibrotic genes [4,5,6] that can slow down these pathological changes. In recent years, microRNA (miR) assessment has been increasingly used to study the molecular genetic pathways of various processes, including the activity of different types of adipose tissue (AT). MiRs are small non-coding RNA (19–25 nucleotides long) that regulate gene expression by affecting the matrix RNA. Over 1500 representatives of this class of molecules have already been described in humans. Changes in their expression are observed in many pathological conditions, and miRs are involved in various biological processes such as apoptosis, proliferation, and differentiation [7]. About 60% of the genome is a target of miRs. The expression of miRs is tissue-specific, but in addition to their effects inside the cell, miRs can reach outside the cell and perform their functions in other cells. They are found in various biological fluids: blood (serum, plasma), urine, saliva, cerebrospinal fluid. Thus, exosomal miRs released from AT can regulate gene expression in other tissues. The interest in AT-produced miRs derives from the fact that the AT is both a miR-producing organ and a target of the miRs it secretes, and that miRs entering the circulation from AT are an important source of miRs in plasma [8,9]. Subcutaneous AT (SAT), due to its location, represents a successful purpose for research on change in expression of miRs in obese patients. Comparison of changes in the level of miRs in SAT and plasma allows one to reveal those miRs which are plasma markers of the processes occurring in SAT. Given the pronounced influence of the AT on the circulatory profile of miRs, studies simultaneously assessing the dynamics of miRs in the AT and plasma are of particular interest to identify those of miRs that change concomitantly. Recent studies have demonstrated that obesity alters the profile of a number of circulating miRs [9,10,11], including miR-142-3p and miR-378 [10,12,13]. Our interest in these miRs stems from their important and ambiguous roles in the pathogenesis of obesity. To identify key miRs and gene markers in adipose tissue, we analyzed the literature data to generate a panel of miRs that have been shown to play a significant role in adipogenesis, inflammation, and fibrosis in the adipose tissue according to other researchers [7,8,9,10,12]. We examined the expression level of selected miRs in SAT from obese patients and compared them with a group of healthy controls. In further analysis, we included those which expression level significantly differed in healthy and obese patients before weight loss and returned to normal values after weight normalization. These were miR-27, miR-93, miR-142, miR-155, and miR-378. Then, we examined the expression level of these miRs in SAT and in plasma from obese patients during sibutramine therapy. Significant changes during sibutramine therapy were found for miR-142 and miR-378. In vitro studies have shown that at the differentiation stage of progenitor cells (3T3-L1 preadipocytes and ST2 mesenchymal progenitor cell lines), the increase in expression of miR-378 stimulates their differentiation into adipocytes [14], the synthesis and accumulation of triglycerides and fatty acids in vacuoles of white adipose tissue (WAT). These effects are implemented by increasing the expression of fatty acid synthase, enlarging the size of lipid droplets and thus stimulating lipogenesis without affecting β-oxidation of fatty acids [15]. In experiments in obese mice, miR-378 is designated as an inhibitor of insulin signaling, by inhibiting p110a, regulator of glucose and fatty acid metabolism, by counteracting peroxisome proliferator-activated receptor gamma coactivator 1-beta (PGC-1β) [16] and is potentially associated with the development of insulin resistance (IR), while other studies have demonstrated its association with activation of adipogenesis of BAT and protection from diet-induced obesity [17,18]. Phosphodiesterase 1b (PDE1b) is a direct target of miR-378 in BAT but not in WAT [12,13]. MiR-378-mediated inhibition of PDE1b regulates the synthesis of cyclic adenosine monophosphate and thus enhances the differentiation of brown adipocytes, increasing the volume of BAT [18]. Furthermore, miR-378 is involved in the regulation of mitochondrial metabolism and energy homeostasis in mice through a PGC-1β-controlled transcriptional network [16]. Thus, data from the literature indicate that miR-378 may play various roles under diverse conditions and in different fat depots (Figure 1A).

Being expressed in the AT, miR-142 regulates lipid metabolism and adipocyte differentiation and reduces triglyceride accumulation in adipocytes by inhibiting fatty acid synthesis gene expression and enhancing fatty acid oxidation gene expression [19]. MiR-142 has been shown to be involved in adipogenesis by inhibiting high-mobility group protein (HMG) A1. Some older studies have linked miR-142 overexpression to inflammation activation, although the genetic pathway has not been established. But recent studies have demonstrated involvement of miR-142 in the inhibition pathway of transforming growth factor beta (TGF-β) and TGF-β receptor, antifibrotic effects and a decrease in type 1 collagen formation with its overexpression [20]. Another mechanism for the antifibrotic effects of miR-142 is the suppression of expression of genes related to the extracellular matrix (COL1A1 encoding the 1-alpha chain of type I collagen and COL3A1 encoding the 1-alpha chain of type III collagen) [21]. Hyperexpression of miR-142 suppresses profibrotic gene expression in cardiomyocytes by targeting HMGB1 [22]. Through similar mechanisms, miR-142 is involved in antifibrotic processes in the liver, its expression is dramatically reduced in liver cirrhosis, and ectopic expression of miR-142 in activated stellate liver cells decreases profibrotic markers [23] (Figure 1B). Moreover, we did not find in the available literature any data on the relationship between miR-142 expression in the AT and fibrosis markers.

The continuing expansion of obesity dictates the need to look for new methods of treatment and its personalization depending on various determinants, using existing approaches to therapy. Therapeutic methods that can reduce fibrosis activation are an attractive option in the treatment of obesity. Given the fact that BAT activation is associated with inhibition of profibrotic genes [4,5,6], these changes may be of a concomitant character. Respectively, increased energy expenditure by activation of BAT is the promising directions in the development of treatment of obesity and associated metabolic disorders [24]. Among the drugs registered in the Russian Federation for the treatment of obesity, such an effect is assumed for the neurotransmitter reuptake inhibitor sibutramine. This drug belongs to a very small group of drugs that, in addition to their central action (serotonin reuptake inhibitor) on appetite, have a peripheral action in the form of catecholaminergic effects, inhibiting norepinephrine (NE) reuptake. NE activates thermogenesis by stimulating β3-adrenoreceptors, increasing energy expenditure. β3-adrenergic stimulation leads to proliferation of preadipocytes through the brown fat cell pathway. This suggests that sibutramine may modulate BAT activity. Previously, the presence of such effects in sibutramine was demonstrated in an experiment [25]. We did not find similar studies for humans in the available literature. In summary, miR-378 has been described as one of three key miRs (miR-143, miR-145 and miR-378) affecting adipocyte differentiation, a positive regulator of adipogenesis in classical BAT, similar to the miR-193a/b cluster and miR-365, miR-182 and miR-203, and it plays a role in the regulation of ADIPOQ transcriptional activity and mitochondrial biogenesis [26]. MiR-142 is a serious candidate to be an inhibitor of fibrogenesis in various tissues [20,21,22,23]. However, under different conditions (plasma or AT, recent obesity or long-term obesity, presence or absence of metabolic disorders), their roles and functions may differ. This leads us to expect that the co-evaluation of their expression and dynamics in obese patients of different severity and duration before and after weight loss on sibutramine therapy will allow us to clarify their effects. In addition, revealing the relationship of their dynamics not only with changes in weight and metabolic status but also with the fact of sibutramine intake may improve our understanding of the mechanisms of action of this drug. Such an analysis is attempted in this article.

## 2. Results

### 2.1. Assessment of Biochemical Markers of Inflammation, Oxidative Stress and Fibrosis, Assessment of Adipocytokine Levels

According to our criteria, the research included 51 patients with obesity: 31.4% with first-degree obesity, 39.2% with second-degree obesity, and 29.4% with third-degree obesity. Laboratory results of patients before treatment and after 6 months of therapy are provided in Table 1.

The average glycaemia level was within normal range, though 13.7% of patients had dysglycaemia. Mean lipids level was also in the range of target values according to stratification of risk; at the same time, 3.9% of patients received therapy with statins. The blood pressure (BP) level was within a normal range and did not differ from that in the healthy control group (HCG) (Table 1), but 7.8% of patients received hypotensive therapy. Significant dynamics of the systolic BP after 6 months of therapy was not noted (118.9 ± 12.6 mm Hg vs. 120.3 ± 9.5 mm Hg, *p* = 0.4); however, a small increase in diastolic BP was shown (71.6 ± 9.1 mm Hg vs. 75.7 ± 6.7 mm Hg, *p* = 0.02). Insulin level and homeostasis model assessment of insulin resistance (HOMA-IR) before treatment corresponded to values of hyperinsulinemia (>109 pmol/L) and insulin resistance (IR) (>2.5); after weight-loss intervention, a significant decrease in these indicators was noted. During weight reduction, we obtained significant positive dynamics of adipokine level: an increase in adiponectin level and a decrease in leptin level, and also a decrease in level of markers of fibrosis—type 1 procollagen C-terminal propeptide (PICP) and galectin-3. At the same time, type 3 procollagen N-terminal propeptide (PIIINP) level increased. Level of inflammation marker C-reactive protein (CRP) did not change significantly. Among oxidative stress markers, a considerable decrease in the myeloperoxidase (MPO) level was revealed (*p* = 0.01), but there were no changes in paraoxonase-1 (PO-1) level (*p* = 0.1).

### 2.2. Assessment of miR-378 and miR-142 Expression in SAT and in Plasma before Sibutramine Treatment

We compared miR-378 and miR-142 expression in SAT in patients with obesity (*n* = 32) and HCG (*n* = 10) who underwent a SAT biopsy. During the research, we found that miR-378 expression in SAT in patients with obesity did not significantly differ from that in healthy participants, though there was a tendency to increase (Figure 1A). At the same time, miR-142 expression in SAT in the patients with obesity was significantly lower (*p* = 0.04) than in the HCG (Figure 2A).

Based on a comparison of miR-378 and miR-142 expression in SAT and in plasma in patients with obesity (*n* = 51) and in the HCG (*n* = 10), miR-378 expression in plasma in the obese patients did not significantly differ from that in the healthy subjects; at the same time, we found a tendency to increase, similar to that in SAT (Figure 1B). Thus, changes in miR-378 in SAT and in plasma in obesity had a unidirectional character. MiR-142 level in plasma of patients with obesity had boundary dynamics (*p* = 0.05) with a tendency to increase (Figure 2B). Thus, changes in miR-142 expression in SAT and in plasma at the development of obesity had a multidirectional character.

Significant differences in expression of both miR-378 and miR-142 in SAT and in plasma in healthy people were not revealed.

We also analyzed factors which could be associated with expression of the studied miRs in SAT. We first considered such factors as degree and duration of obesity.

We revealed significant differences in miR-378 expression depending on the obesity degree and duration. Thus, in the first degree of obesity, the miR-378 expression was significantly raised in SAT in comparison with the HCG (*p* = 0.05), and in higher degrees of obesity, the distinction was statistically insignificant, though a tendency toward increased expression remained. In the same way, in obesity experienced for less than 10 years, the miR-378 expression was significantly raised, while with longer duration of obesity, the distinction was statistically insignificant (Figure 3A,C).

When analyzing the expression of miR-142, depending on the severity of obesity, we found no significant changes in the level of miR-142 in SAT with different degrees of obesity compared to the HCG (Figure 3B). The relationship with the duration of obesity was as follows: the level of miR-142 was significantly lower compared to HCG with a long duration of obesity (more than 10 years) (*p* < 0.02), and with a duration of obesity of 5–10 years, no significant decrease in the level of miR-142 was detected (*p* > 0.05) (Figure 3D).

### 2.3. Influence of Sibutramine Treatment on Dynamics of miR-378 and miR-142 Expression in Plasma and in SAT

In 13 patients with obesity, after 6 months of sibutramine therapy, the repeated SAT biopsy was taken. Blood sampling for the assessment of the studied miRs in plasma was carried out in 33 patients with obesity.

During sibutramine therapy, expression of miR-378 in SAT tended to increase but did not reach statistical significance, while in plasma, a reliable increase in the miR-378 level was revealed (*p* = 0.01) (Figure 1C,D). However, when comparing all patients with obesity before treatment and the patients receiving sibutramine for 6 months, a reliable elevation in the expression level of miR-378 in SAT was revealed; other differences in comparison with the previous selection were not observed. Thus, it is again possible to conclude that there were consensual changes in the expression of miR-378 in plasma and in SAT.

MiR-142 expression, by contrast, significantly increased in SAT of patients with obesity in 6 months of sibutramine therapy (*p* = 0.001) (Figure 2C). The miR-142 level in plasma did not significantly change during therapy (Figure 2D), though it also showed a tendency to increase.

Depending on efficiency of therapy, all patients were divided on responders and non-responders. In the group of responders, the SAT biopsy was taken for nine people; in the non-responders group, it was taken for four people. When comparing miR-378 expression before therapy in groups of responders and non-responders, significant differences were revealed: in the group of responders, the miR-378 expression was lower than in the non-responders group (*p* = 0.05) (Figure 4A). When comparing the expression of miR-378 in plasma in responders and non-responders, no significant differences were obtained (Figure 4B). By comparison of miR-378 expression in SAT in responders and non-responders prior to therapy with the HCG, we found that in responders, it was similar to that in the HCG, while in non-responders, there was a tendency toward higher values.

When comparing miR-142 expression in responders and in non-responders in SAT and in plasma, significant distinctions were also revealed, but they had an opposite character: in responders, miR-142 expression in SAT was lower than in non-responders (*p* = 0.01); miR-142 expression in SAT in non-responders did not differ from that in the HCG (*p* = 0.2). At the same time, in plasma, the miR-142 level was higher than in non-responders (*p* = 0.03) (Figure 4C,D).

To specify the key determinant of miR-378 and miR-142 expression changes during therapy (weight reduction or sibutramine therapy), we compared the expression of the studied miRs in responders and non-responders at the end of treatment. There were no significant differences in the expression of both miR-142 (*p* = 0.5) and miR-378 (*p* = 0.7) in SAT in responders and non-responders, indicating a relationship of expression changes specifically with sibutramine therapy.

### 2.4. Comparison of the miR-378 and miR-142 Levels in Plasma and in SAT of Patients with Obesity with Metabolic Parameters and Markers of Fibrosis

As described above, a significant decrease in leptin, galectin-3, and PICP levels and an increase in adiponectin and PIIINP levels were found during therapy. We analyzed the relationships between the dynamics of these parameters and changes in the expression of the studied miRs. A negative association of miR-142 expression with PICP level before treatment (r = −0.47, *p* = 0.02) was the only significant finding. The absence of correlation of miR-142 expression in plasma with the level of fibrosis markers suggests that this relationship is specific to fibrogenesis in SAT. We attribute the absence of a similar relationship after treatment to the significantly lower number of patients reaching point 2 and the presence of different weight dynamics (responders/non-responders), which did not allow sufficient statistical significance for this relationship at point 2.

## 3. Discussion

Among the many miRs closely related to the functional activity of the AT and dysregulated in obesity [8], two that are related to the efficacy of sibutramine therapy attracted our attention: miR-378 and miR-142. We studied their representation both in the SAT and in the circulation. Given the large volume of AT in obesity and its high secretory activity, it is a significant source of miRs in the circulation [8,9]. Moreover, the significance of the AT contribution to the circulatory profile of miRs, in our opinion, should be evaluated on a case-by-case basis, especially for those miRs that are also highly expressed in other tissues. This characterization applies in particular to miR-378 and miR-142. To determine whether the circulating levels of these miRs reflect changes in their expression in SAT, we performed a parallel evaluation of miR-378 and miR-142 expression in SAT and their levels in plasma, before and during sibutramine therapy.

According to the literature, miR-378 and miR-142 have multidirectional effects on preadipocyte differentiation into adipocytes and adipogenesis of WAT. Increased expression of miR-378 stimulates differentiation of precursor cells into adipocytes and increases triglyceride deposition therein [15,27]. Thus, miR-378 increases both adipocyte hyperplasia and hypertrophy, whereas miR-142 inhibits adipogenic differentiation and autophagy in obesity, targeting Krueppel-like factor 9 [19].

Considering this, the multidirectional character of the changes in the expression of these miRs, which is the most pronounced in the early stages of obesity, seems very logical. The most pronounced elevation in miR-378 expression and reduction in miR-142 expression exactly at the first stage of obesity can be explained by the fact that at the beginning of obesity there is a maximal activation of mechanisms ensuring the expansion of the SAT volume to store the excess incoming energy. Other researchers have also noted that miR-378 expression increases during normal adipogenesis in humans [15]. The subsequent “pseudonormalization” of expression at higher degrees of obesity can be explained by the depletion of adaptive mechanisms. Moreover, as the duration of obesity increases, the significance of increased miR-378 expression is lost, whereas the decrease in miR-142 expression becomes more pronounced. As the duration of obesity grows, proinflammatory cytokines and hypoxia factors activate profibrogenic genes and disrupt the activity of miRs that inhibit fibrosis, among which miR-142 occupies an important place. The decreased expression of miR-378 may also be due to prolonged exposure to high levels of proinflammatory cytokines. Tumor necrosis factor alpha (TNF-α) and interleukin-6 (IL-6) have been shown to increase miR-378 expression at short exposure times, but this effect is lost with increasing exposure duration [28]. The discovery of the role of miR-142 in the modulation of TGF-β activity has prompted the study of the effects of changes in miR-142 expression on fibrosis processes. In recent years, evidence has emerged demonstrating antifibrotic effects of increased miR-142 expression in various tissues: in scleral fibroblasts [20], in liver star cells (low miR-142 expression was associated with liver cirrhosis) [23], in lungs [29] and cardiomyocytes [22]. Activation of oxidative stress and fibrosis was confirmed in our study by a significant increase in MPO, galectin-3, and PICP levels, and the relationship between miR-142 expression activity in SAT and fibrosis activity was confirmed by a negative correlation between miR-142 expression in SAT and circulating PICP levels. Our results are in agreement with those of Li et al. [20], who demonstrated that overexpression of miR-142 in scleral fibroblasts reduces TGF-β1 and collagen I expression. A series of studies on pulmonary fibrosis showed the importance of miR determination in plasma, in the miR source organ, and in the target organ. For example, when miR-142-3p was assessed in plasma and saliva of patients with pulmonary fibrosis [21,30,31], its expression was elevated, whereas low miR-142 expression was determined in lung tissue. In these studies, there is a clear analogy with our results, demonstrating decreased expression of miR-142 in SAT in obesity compared to healthy subjects and increased levels of miR-142 in plasma of obese patients.

Circulating miRs are valuable biomarkers of systemic diseases and potential therapeutic targets because of their ease of assessment. Several studies have examined the pattern of miRs in circulation in obesity and their changes after weight loss [10]. Our findings regarding changes in miR-142 levels are similar to those of Ortega et al., who studied the circulating profile of miRs in 36 men with morbid obesity before and after bariatric treatment [10]. As in our work, this study showed a marked increase in miR-142-3p (*p* < 0.0001) in the circulation. At the same time, surgery-induced (but not diet-induced) weight loss led to a significant decrease in several other miRs but not miR-142 in the circulation. Importantly, this study also assessed levels of transforming growth factor receptor (TGFR), for which circulating levels were associated with those of miR-142-3p. The key targets for miR-142 identified in silico are the transforming growth factor (TGF) beta receptor 1 gene and the leukemia inhibitory factor-α (LIF-α) receptor gene. Members of the TGF-superfamily [32], like LIF [33], are known to regulate many aspects of adipocyte development and are involved in the development of obesity and the regulation of energy expenditure. In another study evaluating the circulatory profile of miRs in obese adolescents, there was also a significant increase in miR-142-3p [32].

The authors of that study noted that miR-142 levels were closely related to body mass index (BMI), adiponectin, leptin, insulin, HOMA-IR, and lipid level in plasma. We did not observe such multiple relationships with metabolic parameters, which may be explained by the lower severity of obesity in the patients in our study and by the fact that some patients received hypolipidemic therapy and the obesity drug. Similarly, another study, also demonstrating increased levels of miR-142 in the circulation, did not find its association with levels of inflammatory markers (CRP, IL-6, TNF-a) in the circulation.

Thus, miR-142 showed consistently elevated levels in the circulation in obesity in multiple studies, which may reflect an attempt to counteract the pro-inflammatory and profibrotic status characterizing obesity [32,34]. In contrast, miR-378 demonstrated a clear concordance between the changes of expression in SAT and the level in the circulation, indicating that this miR is a good circulatory biomarker, reflecting changes in the activity of several biological processes (adipogenesis, browning, lipolysis) in SAT [15,18].

Thus, a feature of our study was the simultaneous assessment of the studied miRs in the SAT and circulation, which allowed us to clarify whether the change in miR levels in the circulation reflects dysregulation of their expression in the AT or systemic changes in obesity. Studies with simultaneous assessment of miRs in the circulation and AT are extremely scarce in the literature. This is supported by a recent meta-analysis [33] that included 17 studies evaluating the dynamics of miR expression before and after surgical treatment of obesity. Only three of these evaluated miR expression in SAT and not a single one showed a parallel evaluation. This meta-analysis reported that both miR-142 and miR-378 were noted to be differentially expressed in plasma of obese and normal-weight individuals. But, while the results for the direction of change of miR-142 were consistent in the studies presented [10,35], decreasing in the postoperative period, the results for miR-378 were divergent. In a study by Alkandari et al. [36] miR-378 was elevated in plasma before treatment and its level decreased after surgery, while in a study by Lirun et al. [37], its level was decreased before surgery and increased after. We observed the dynamics of miR-378 levels in plasma similar to those of Lirun et al. [37].

The next aspect of our study was to assess the dynamics of miR-378 and miR-142 during sibutramine therapy. Changes in miR expression can be a useful tool to clarify the mechanisms of drug action. Thus, Xue et al. [28] showed a significant increase in miR-378 expression under the influence of rosiglitazone, demonstrating the involvement of the adipocyte differentiation and adipogenesis pathways regulated by it in the mechanisms of action of this drug. Given the known data on the action of sibutramine, including catecholaminergic effects, we expected a more pronounced change in miR-378 expression during therapy. We observed a significant increase in its level in the circulation but only an increasing trend in SAT, which may reflect the systemic action of the drug. Moreover, miR-378 overexpression is associated with the activation of adipogenesis of classical BAT [18], localized mainly in the supraclavicular region rather than with the browning of SAT in the abdominal region, from where SAT was sampled. Classical BAT is extremely rare and sparsely represented in obese subjects and is unlikely to contribute to changes in circulatory levels of miR-378, which may partially explain our results. At the same time, the expression of miR-142 significantly increased during therapy in SAT but not in plasma, which may indicate the ability of sibutramine to reduce the activity of profibrotic genes and fibrogenesis processes in SAT. This may have important implications for the maintenance of SAT plasticity and metabolic health in obesity. The fact that the expression changes are associated specifically with sibutramine therapy rather than with BMI reduction during therapy is confirmed by the fact that there was no difference in the expression of the studied miRs after therapy in responders and non-responders.

Another interesting finding of our study was the differences in miR-378 and miR-142 expression in patients with good response to sibutramine therapy and in non-responders. Good response to therapy was characterized by those patients in whom miR-378 expression was not elevated (significantly lower than in non-responders), and in contrast, miR-142 expression had a marked decrease. These patients seem to have a point of application for therapy, whereas non-responders, who had no significant deviations in expression from healthy individuals’ parameters, did not. Attempts to use changes in miR expression to predict response to interventions have been made before. For example, in 2013, Milagro et al. studied the expression of miRs in peripheral blood leukocytes of 10 obese women, during an 8-week hypocaloric diet. Patients were divided into responders (>5% weight loss, *n* = 5) and non-responders (<5% weight loss, *n* = 5) [38], the same as in our study. Differences in the expression of the five miRs before intervention between the two groups were found. In the non-responder group, miR-935 and miR-4772 were up-regulated, whereas miR-223, 224, and 376b were down-regulated [38]. Marques-Rocha et al., studied the modulation of the expression of nine inflammation-related miRs in leukocytes in MS after an 8-week hypocaloric diet and found a significant change in two miRs after the intervention: miR-155-3p was strongly decreased and miR-7b was significantly up-regulated [39]. Several studies have found differences in miR profiles depending on the type of diet [40,41]. In a study by Assmann et al., among the miRs that differed in circulation in obese patients and were significantly associated with response to a low-fat diet was miR-142 [41], and in a study by Giardina et al., miR-378 was among the 13 miRs whose expression was reduced in responders compared with non-responders on both a low-carb and a low-fat diet [40]. These data on the one hand confirm the importance of the mechanisms determined by these miRs for treatment efficacy but on the other hand require consideration of dietary characteristics. To exclude the influence of dietary characteristics on miR expression changes, all patients in our study were given the same type of dietary and physical activity recommendations, which were strictly monitored.

We found no data on prediction of response to drug therapy for obesity in the available literature. Drug therapy requires high-precision response prediction due to the fact that the fraction of responders undergoing therapy is about 60%, and the cost is quite high. Thus, according to Formichi et al., 19.2% of patients dropped out of the study of glucagon-like peptide 1 receptor agonists (GLP1ra) therapy because of side effects [42]. According to our data in real clinical practice, 46.1% of patients fully received a 6-month course of sibutramine therapy (20.5% dropped out due to side effects, 33.4%—due to unsatisfactory price/effect ratio), and 48% of patients received GLP1ra liraglutide (16% of patients dropped out due to side effects, 36%—due to unsatisfactory price/effect ratio) [43]. In a study by Formichi et al. [42], as in our study, a block of miRs involved in metabolism processes, not AT but glucose and differing in expression levels in diabetes mellitus type 2 with obesity, was selected based on literature data and the miR dynamics in plasma during GLP1ra therapy was evaluated. Given that GLP1ra are also registered as drugs for the treatment of obesity, and considering the significant weight reduction according to this study (*p* < 0.05), we present it as an example of an attempt to predict the response to drug therapy for weight loss based on the miRs’ profile. The study analyzed the dynamics of 8 miRs, including miR-378, in circulation, and showed that high baseline levels of miR-375 had the greatest predictive value for achieving a good glycemic effect. However, baseline levels of miR-378-3p also correlated with a reduction in HbA1c by 12 months relative to baseline (*p* = 0.002). The effects on weight loss were analyzed separately. As in our study, responders were defined as those who reduced weight by 5% or more. Responders had significantly higher baseline levels of miR-15a-5p than non-responders (*p* = 0.03). These results suggest the need to account for glycemic abnormalities when analyzing miR-378 changes. In the available literature, we did not find any studies on the dynamics of miR expression in SAT and plasma in obese patients treated with sibutramine or similar drugs (serotonin receptor agonists), which determines the novelty of our findings on miR-378 and miR-142 expression changes in such patients.

## 4. Materials and Methods

### 4.1. Clinical Characteristics of the Examined Patients, Study Design

This study included obese patients who met the following inclusion criteria:Male or female;Age 18–56;BMI > 30 kg/m^2^;Absence of secondary causes of obesity;Consent to participate in the study and take the drug as recommended.

Separate informed consent was signed for the SAT biopsy.

Taking into account catecholaminergic-associated adverse reactions to sibutramine, the inclusion criteria were cardiovascular diseases (ischemic heart disease, congestive heart failure, heart defects, acute myocardial infarction, acute stroke, arrhythmia, tachycardia), uncontrolled arterial hypertension, as well as individual intolerance reactions, anorexia or bulimia, thyrotoxicosis, liver or kidney failure, benign prostatic hyperplasia or glaucoma.

In accordance with these criteria, 64 patients with varying degrees of obesity were included in the study; however, due to the poor quality of the samples, 51 patients were included in the final analysis (BMI 37.3 ± 4.8 (33.8; 40.2) kg/m^2^, mean age 37.5 ± 10.8 years), and 10 healthy non-obese volunteers (BMI 22.4 ± 1.5 (21.0; 23.6) kg/m^2^), comparable by age (40.4 ± 9.5 years) and sex ratio, constituted the control group.

Informed consent to perform SAT biopsy was signed by 32 obese patients: the first SAT sampling (before therapy) was performed in all 32 patients, repeated biopsy (after 6 months of sibutramine therapy)—in 13 patients who followed all treatment recommendations for 6 months. Informed consent for SAT biopsy was signed by all 10 subjects from the control group, who also underwent SAT sampling.

The study included two visits, the first one (V1) before the start of sibutramine therapy and the second one (V2) 6 months after initiation of the therapy. At visit V1, after signing the informed consent, the patients underwent an examination that included taking an anamnesis, assessment of anthropometric parameters (weight, height, calculating BMI using the formula: body weight (kg)/height^2^ (m)), assessment of the level of arterial BP (Table 2).

To clarify the reasons for the expression dynamics of the studied miRs (weight loss or drug effect), we compared their dynamics in patients who lost weight (responders) and those who did not lose weight (non-responders). Responders were considered patients whose body weight loss reached 5% or more by the end of the third month of therapy, with subsequent maintenance or increase in weight loss by the sixth month of therapy.

Taking into account the possible influence of nutritional components on the miR profile [44], after a basal examination, obese patients were given standardized lifestyle recommendations: moderate hypocaloric nutrition with a calorie deficit (300–500 kcal), balanced in the intake of carbohydrates, proteins and fats (carbohydrates 45–55%, proteins 15–20%, fats 20–35%) with the exclusion of easily digestible carbohydrates and the restriction of animal fats (no more than 10%) of total calories and a high intake of coarse fibers (30 g/day), dosed physical activity. Sibutramine was prescribed at a dosage of 10 mg/day (1 capsule 30 min before meals).

Of the laboratory parameters, the following were evaluated: glucose, lipid metabolism, levels of hormones involved in the regulation of fat metabolism (leptin, adiponectin, insulin), HOMA-IR, markers of inflammation (CRP), oxidative stress (MPO and PO-1) and fibrosis (PICP, PIIINP and galectin-3). A similar scope of examination was performed on V2.

### 4.2. Laboratory Methods

Biochemical parameters were assessed using a Cobas c311 automated biochemistry analyzer (Roche, Basel, Switzerland) and commercial kits (Roche reagent kits, Basel, Switzerland). Reference values for various biochemical parameters: fasting plasma glucose 3.30–6.10 mmol/L (measurement range 0.11–41.1 mmol/L); total cholesterol (TC) (measurement range 0.1–20.7 mmol/L, normal values 3.50–5.00 mmol/L); serum triglycerides (measurement range 0.1–10.0 mmol/L, normal values < 1.77 mmol/L); high-density lipoprotein cholesterol (HDLC) (measurement range 0.08–3.12 mmol/L, normal values for females > 1.2 mmol/L, for males > 1.0 mmol/L). Serum insulin levels were measured using a Cobas e411 fully automated immunochemical electrochemiluminescence analyzer (Roche, Basel, Switzerland) and commercial Cobas Insulin Elecsys kits (Roche, Basel, Switzerland) within a measuring range of 1.39–6945 IU/mL and a normal value of 17.8–173.0 pmol/L. The pmol/L to μU/mL conversion factor was 0.144

The HOMA index (insulin resistance index) was used to calculate the formula:$$\text{HOMA-IR}=\frac{Glucose\left(\frac{mmol}{L}\right)\times Insulin(\frac{\mu IU}{mL})}{22.5}$$

The levels of leptin in blood serum and adiponectin in blood plasma were assessed by enzyme immunoassay using an automatic analyzer (Bio-Rad 680—monometer, automated analyzer, Bio-Rad, Hercules, CA, USA). For adiponectin (commercial BioVendor kit, Brno, Czech Republic), measurement range 0.026–100 μg/mL, sensitivity coefficient 0.026 μg/mL, for leptin (commercial DBC kit for enzyme immunoassay (enzyme-linked immunosorbent assay)), measurement range from 2.0 to 11.0 ng/mL, sensitivity 0.5 ng/mL. Serum levels of galectin-3 (R&D system, Minneapolis, MN, USA), PICP (USCN Life Science, Wuhan, China), PIIINP (USCN Life Science, Wuhan, China) were assessed by enzyme immunoassay. The study of the level of myeloperoxidase (enzyme-linked immunosorbent assay) was performed with the Human Myeloperoxidase LotP203356 reagent kit, R&DSystems, Minneapolis, MN, USA ng/mL, PO-1 with a Paraoxonase 1 (PON1) reagent kit, LotL190801668, MyBioSource, MyBioSource, San Diego, CA, USA, μg/mL. CRP was assessed using an automatic analyzer (Cobas c311, Roche Diagnostics GmbH, Mannheim, Germany) and commercial kits (reagent kits, Roche, Basel, Switzerland).

### 4.3. Biopsy of Subcutaneous Adipose Tissue

Biopsy of subcutaneous adipose tissue was carried out by puncture of subcutaneous fat in the area of the anterior abdominal wall. The puncture was performed in the morning, in the fasting state, in the supine position using conventional injection needles according to the “free hand” method. Obtaining aspiration material was carried out without anesthesia, with preliminary treatment of the surgical field with 70% alcohol. The puncture was made in the umbilical region 2–3 cm to the right and below the navel, then the subcutaneous adipose tissue was aspirated with traction movements, with a total volume of not more than 2 mL. After that, for transportation to the biobanking site, the Eppendorfs were immediately placed in liquid nitrogen. Biosamples were stored at a temperature of −80 °C.

### 4.4. RNA Isolation and miR Quantification

Total RNA was isolated from adipose tissue biopsies and plasma using the ExtractRNA reagent for the isolation of total RNA from biological samples (Evrogen, cat. no. BC032, Moscow, Russia). Reverse transcription was performed using the TaqMan™ MicroRNA Reverse Transcription Kit (Thermo Fisher Scientific, Waltham, MA, USA) according to the manufacturer’s recommendations. The reaction mixture (5 μL total volume) contained 2.33 μL master mix (0.05 μL dNTP Mix 100 mM; 0.33 μL Multi Scribe reverse transcriptase (50 U/μL); 0.063 μL RNase inhibitor 20 units/μL; 0.5 μL 10× RT buffer, 1.386 μL deionized water), 1.66 μL RNA solution and 1 μL miR-specific primer.

In our work, expression levels of the following miRs were analyzed: hsa-miR-378a-3p, hsa-miR-142-3p. The miR reverse-transcription reaction was performed using a 96-well Veriti thermal cycler (Applied Biosystems, Foster City, CA, USA), in which the samples were incubated at 16 °C for 30 min, then 42 °C for 30 min and 85 °C for 5 min.

MiR amplification by qPCR was performed using TaqMan Universal Master Mix II (Life Technologies, Carlsbad, CA, USA; cat. no. 4440040) and the following TaqMan MicroRNA Assays were used to detect microRNAs of interest, namely hsa-miR-378a-3p (Assay ID 000464) and hsa-miR-142a-3p (Assay ID 001314), according to the manufacturer’s recommendation.

To determine the operating range of the detection system for miR analysis, a standard curve was constructed (a plot of threshold cycles versus the logarithm of concentrations in a series of sample dilutions). Synthetic oligoribonucleotides (Synthol) with a known concentration, identical to the analyzed miR, were used to construct a standard curve.

### 4.5. Statistical Methods

Statistical analysis was performed using STATISTICA 10 (StatSoft Inc., Tulsa, OK, USA) for Windows. Data were presented as mean ± standard deviation. The distribution of the studied variables deviated from normal (normality test was calculated using the Kolmogorov–Smirnov criterion, dispersion was calculated using the Levene’s test). Due to the fact that the distribution in the groups for some parameters is unequal, the use of simple methods of calculation (Student’s *t*-criterion, etc.) is inappropriate. Therefore, we used non-parametric statistical methods to evaluate the studied groups (Mann–Whitney test was used to compare two independent samples with an interval scale, Wilcoxon *t*-criterion was used to evaluate the relationship between a factor and a dependent variable, to compare two dependent samples with each other by the level of expression of a feature). The power of the study for 50 patients was 0.8. The critical significance level (*p*) for testing statistical hypotheses when comparing statistical indicators was less than 0.05.

## 5. Conclusions

In this study, we evaluated the expression of miRs 378 and 142 in SAT and their plasma levels in obesity of different degrees and duration and then their dynamics during weight loss on sibutramine therapy. Currently, miR-378 can be considered one of the key miRs in the metabolism of the AT, as it affects adipocyte differentiation, adipogenesis in both white and classical BAT, mitochondrial energetics and adipocytokine production [26], thus regulating several metabolic pathways in AT. In our work, we showed that its expression depends on determinants such as duration and severity of obesity. We did not observe a statistically significant increase in miR-378 expression in SAT during the 6-month sibutramine therapy, but a more pronounced increase in its plasma levels suggests the potential for activation of classical BAT. We hope to obtain a confirmation of this hypothesis in our future studies. Altered expression of miR-142 in obesity has been reported in a small number of studies, but given that its main target is the inhibition of the TGF-β/Smad pathway, which is involved in the regulation of WAT adipogenesis and the development of its dysfunction by enhancing the expression of fibroblast signature genes, inhibition of BAT adipogenesis [45], it can be considered as a candidate for antifibrogenic miRs, which are important for maintaining the metabolic health of BAT [45]. However, its importance remains to be elucidated. The ability of sibutramine to alter miR-142 expression in SAT was an important finding of our study. Both studied miRs were shown to be potential biomarkers for predicting the efficacy of sibutramine treatment.

Limitations of the study. Limitations include the small study size, lack of experimental molecular genetic studies confirming the drug’s contribution to the activation of determined miRs 378 and 142 genes specifically through these miRs. We also did not have a group of obese patients who lost weight based on lifestyle changes without the addition of drug therapy, which would have been useful to confirm the association of the detected changes specifically with drug intake. Meanwhile, given the dynamics of miRs 378 and 142, with no clear relationship with BMI dynamics, our assumption of this relationship seems to be reasonable.

## Data Availability

Data can be provided upon request.

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
