# Peer review of "Tissue and Circulating MicroRNAs 378 and 142 as Biomarkers of Obesity and Its Treatment Response"

_ijms, 2023, doi:10.3390/ijms241713426_

Round 1

Reviewer 1 Report

Georgy Matveev and the group discuss the role of “Some tissue and circulating microRNAs as biomarkers of obesity and of response to obesity treatment”. In their study, they targeted two microRNA- hsa-miR-378a-3p and hsa-miR-142-3p and found that a maximal increase in the expression of the adipogenesis inducer miR378 in the early stages of obesity, a progressive decrease in the expression of the fibrogenesis inhibitor miR142 in SAT with the growth of duration of obesity and the likely presence of antifibrogenic effects of sibutramine realized through miR142 activation. This miRNA-142-3p could be a possible biomarker for obesity diagnosis. 

The work performed by authors is significant and has the potential to be published in the journal. However, the article can be improved by addressing the following comments and suggestions. 

Scientific Comments

  1. Two times “of : not needed in the title of the manuscript, kindly modify it
  2. The authors need to move the lines of microRNA at the starting paragraph of the introduction section so that reader could easily understand the point that the authors want to make from this study
  3. Please modify the table in a professional way including the statistical part shown in the table, the authors could check it from previous MDPI journals
  4. Line 224-225 please modify the sentences, there should be no question mark in the lines
  5. Line 245-246, no need to add references in the first paragraph of the discussion

6.   Did the authors calculate the power of the study? If yes, please mention it in the revised MS.

7.   What are the main selection criteria for hsa-miR-378a-3p and hsa-miR-142-3p? Discuss in revised MS. 

8. If possible, explain the role of miR-378a and miR-142 in obesity by a figure or in tabular form. 

9. Line 71-72, Recent studies have ……… circulating miRs. Mention the name of the major miRNA that alters/regulated obesity.

10. Line 75-80, you have mentioned the role of miR-378a in obesity with 2-3 references. Please add some more specific references showing the potential role of miR-378a regulation in obesity.

11. Line 316-318, give 1-2 references for this. 

12. Line 335-336 where were the references, kindly add up.

13. According to your experimental result, which one is a better biomarker among both miRNAs for easy and timely diagnosis and further treatment? Mention in conclusion. 

14. The * should be “X” in the definition of HOMA IR

15. All references should be modified based on the MDPI GUIDELINES. 

Minor 

  1. Line 40, “HE activates thermogenesis.”, it should be NE. 
  2. Check the reference pattern. The year should be in bold font. 
  3. Check line 116, is it correct?
  4. Several typo errors are present such as lines 387, 308, and many more.

English editing is required.

Author Response

Dear reviewer!

The team of authors sincerely thanks you for the time you have spent to our work, for your thoughtful and thorough analysis and valuable comments.

We have revised the text in accordance with your comments, highlighting the changes in red font. The corrected typos are highlighted in yellow.

In accordance with the question in 7 «What are the main criteria for selecting has-miR-378a-3p and has-miR-142-3p?», we would like to point out that the data presented are part of a large study evaluating miR dynamics under the influence of different research methods. We have included in the introduction of the article information about the methodology of the study as a whole and why this piece of research focuses on has-miR-378a-3p and has-miR-142-3p.

We are very grateful for the idea presented in paragraph 8 to explain the role of miR-378a and miR-142 in obesity with a figure. Indeed, this is a very illustrative method and we tried to do so by adding Figure 1 to the introduction.

Unfortunately, we found it difficult to fulfill your recommendation in item 9. Line 71-72, «…Mention the name of the major miRNA that alters/regulates obesity.»

The problem is that several dozen microRNAs have now been described that are regulators of adipogenesis, adipose tissue browning, and thermogenesis that are also involved in other aspects of adipocyte physiology, including lipolysis, lipogenesis and lipid droplet formation, glucose uptake, insulin sensitivity, and adipokine secretion.  Therefore, it is not possible to identify any single microRNA as key. Our work has examined selected aspects of miRNA 142 and 378 activity, clarifying the factors affecting their expression in obesity and their role in the dynamics of adipose tissue activity. In terms of the significance of these two miRs in the pathogenesis of obesity, we can say that for miR378, its significant role in the metabolic activity of adipose tissue has been reliably established, which is confirmed by numerous studies by different authors. The alteration of miR142 expression in adipose tissue and plasma in obesity has not been established recently, there are few works on its activity and its significance remains to be clarified. We have added information about it to the discussion and conclusion as suggested by you in item 13.

Let me thank you again for your extremely valuable recommendations, which we hope have significantly improved our work.

Reviewer 2 Report

I reviewed the Manuscript ID  ijms-2516187: “Some tissue and circulating microRNAs as biomarkers of obesity and of response to obesity treatment”.

The subject of the paper is interesting, although I recommend for acceptance after major revision.

 The Authors’ aim was to assess the expression levels of miRs (hsa-miR-378a-3p, hsa-miR-142-3p) in subcutaneous adipose tissue (SAT) and in plasma in patients with different degrees and duration of obesity and during sibutramine therapy. The study included 51 obese patients and 10 healthy normal weight subjects, as control group.

 The study found that before treatment, obese patients had no significant difference in the expression level of miR378 in SAT and plasma compared to the control group, while the expression 19 of miR142 was significantly decreased in SAT and increased in plasma. The degree of obesity and its duration seem to influence the expression level of miR378.

 First the title should be improved.

Overall, the quality of the work is quite good.  Authors described the data with discrete methodology, but without an appropriate description of the statistical methods used.

In the Materials and Methods section, the “Statistical analysis” paragraph is missing. The description of the statistical significance in the tables and in the figures captions is missing.

In my opinion, the paragraph 6 “Patients” (Material and Methods) should become paragraph 1. The “Conclusions” chapter should be placed at the end of the manuscript before “Author contributions”.

Introduction section is described in detailed manner, however the sentence from line 109, to line 113 is poor written, it should be improved.

In the table 1 the number of patients with obesity is 56, while in the text is 51 (line 511).  In this sample taken into consideration for the study there is a large preponderance of female subjects (40 vs 11 male)…

Moreover, in the table1 the SBP and DBP values are missing

I noticed that many acronyms are not explicit in the text, mainly in the Introduction section.

There are different typing errors in the text.

Author Response

Dear  reviewer!

The team of authors sincerely thanks you for the time you devoted to our work, for your thoughtful and thorough analysis and valuable comments.

We have revised the text in accordance with your comments, highlighting the changes in lilac font. Corrected typos are highlighted in yellow background.

We hope that we have been able to improve the title of the article in accordance with your recommendation.

We have supplemented the Materials and Methods section with "Statistical Methods" as per your recommendation

We tried to improve the paragraph (lines 109-113 in the original version) and we really hope we succeeded.

We have supplemented Table 1 and checked the correctness of using abbreviations in the text.

Let me thank you once again for your extremely valuable recommendations, thanks to which we hope we were able to significantly improve our work.

Round 2

Reviewer 1 Report

Reviewer Comments

In the present article “Tissue and circulating microRNAs 378 and 142 as biomarkers of obesity and its treatment response,” the authors have discussed the role of miRNA as a biomarker and therapeutic target for Obesity.  

The authors have done most of the changes as suggested previously. However, there are huge mistakes in the revised file. The paper needs to be read and correct carefully.

Comments

1.      Line 6 3…..[12, 22, &], check this highlighted part.

2.      Line 63, [34,@,%], check this highlighted part.

3.      Line 66, other researchers (), is you forget to cite any reference in this empty bracket?

4.      Line 76, adipocytes [$], check this highlighted part.

5.      Line 83, (PGC-1β)[#] , check this highlighted part.

6.      Line 87,  WAT[@,%], check this highlighted part.

7.      Line 91, PGC-1β-controlled transcriptional network[#], check this highlighted part.

8.      Line 93 (pic.1a). Is it figure 1a? Check and write properly.

9.      Line110 [20](pic.1b). Is it figure 1b? Check and write properly.

10.  Line 132-133, (Arner P, Kulyté A. MicroRNA 132 regulatory networks in human adipose tissue and obesity. Nat Rev Endocrinol. 133 2015;11(5):276-288. doi:10.1038/nrendo.2015.25), check the citation and properly insert.

11.  Line 135, ( ссылка из ref), check the citation and properly insert.

12.  Line 141-142, An attempt to make such an analysis is made in this article, is this sentence is complete?

13.  Check the reference pattern. Revised the reference according IJMS pattern.

Minor English editing is required. 

Author Response

Dear reviewer!

We sincerely  apologise, but we made a mistake when uploading the PDF version of the file. In the Word version there are no defects noted by you and it is the correct version. We resubmitted the version of PDF, having replaced it with the correct one.

Thank you for your attention!